# Trends in progress test performance of medical students at a university in Peru

**Franco Romaní-Romaní***, **César Gutiérrez**

Facultad de Medicina Humana, Universidad de Piura, Lima, Perú

* Franco.romani@udep.edu.pe

## Abstract

### Background

Progress testing is a longitudinal assessment method used to monitor the acquisition and retention of knowledge throughout medical training. While progress tests (PTs) have been widely adopted internationally through collaborative networks of medical schools, in Peru, their implementation has been primarily institutional. This study aimed to evaluate longitudinal trends in PT scores at a Peruvian medical school.

### Methods

We conducted a longitudinal analysis using data from PTs administered annually between 2017 and 2024. The PT assessed students' knowledge based on the subjects completed at the time of testing. Scores ranged from 0 to 250 and were converted to a 20-point scale. Independent variables included number of PTs taken (1–7), year of entry into medical school (entry cohort; 2017–2024), year of test administration (2017–2024), and sex. Generalized estimating equations (GEE) were used to assess score trends over time, applying an identity link function with a Gaussian distribution and robust standard errors clustered by student ID.

### Results

We included 1,899 test scores from 669 medical students. The mean score across all tests was 9.19 (standard deviation = 2.34). No consistent upward trend in PT scores was observed over the study period; scores decreased by 0.088 points per additional year (CI95% CI: −0.147 to −0.029, p = 0.003). Students who completed five PTs scored significantly higher than those who took four (β = 1.40; 95% CI: 0.79 to 2.01). When stratified by entry cohort, no sustained improvement in scores was observed within cohorts over time.

**Data availability statement:** All relevant data are within the manuscript and its Supporting Information files.

**Funding:** Universidad de Piura (PI2501). The funders had no role in study design, data collection and analysis, decision to publish, or preparation of the manuscript.

**Competing interests:** The authors have declared that no competing interests exist.

## Conclusion

Over an eight-year period of administering a progress test at a Peruvian medical school, student performance remained stable, with an average of approximately 50% of questions answered correctly per test. Longitudinal analysis did not reveal a sustained increase in scores as students advanced through the curriculum. This pattern may be explained by the PT design, which assesses only the content covered by students at the time of each administration, in contrast to other PTs that measure end-of-curriculum knowledge across all cohorts. Nevertheless, an increase in median scores was observed during the transition from basic science to clinical subjects.

## Introduction

In medical schools, progress tests (PTs) are longitudinal assessments designed to track students' learning trajectories over time [1,2]. These evaluations provide individualized feedback on knowledge gaps based on each student's progression through the curriculum. PTs emerged from the need to comprehensively assess learning outcomes and evaluate the effectiveness of problem-based curricula. As a result, medical schools began adopting this strategy in the late 1970s [1]. The first implementations started in the United States (University of Missouri–Kansas City), the Netherlands (Maastricht University), and Canada (McMaster University) [3,4]. In subsequent decades, the approach spread across Europe. In South America, Brazilian universities began applying PTs in the late 1990s [5], while adoption in the Middle East began in 2012 [6].

At the international level, progress tests are primarily developed by consortia of medical schools [7–10]. However, some institutions have implemented them as independent initiatives. In general, PTs allow for the assessment of students at multiple points throughout their educational journey. They may be administered three to four times per year and across all years of study, although their mandatory nature varies by institution [11]. Each test is generally cross-sectional in nature, as all students are expected to take the test simultaneously on the same day [12].

PTs are grounded in the principle that students are assessed based on the level of knowledge expected at the completion of their medical education. However, some variations of PTs adjust the expected knowledge level according to the student's stage in the curriculum at the time of testing [13]. The scores obtained from PTs provide valuable data for evaluating the effectiveness of the curriculum in promoting progressive learning, as well as for identifying opportunities for curricular improvement [14]. Longitudinal studies conducted among medical students have consistently demonstrated a progressive increase in PT scores over the course of their education [1,10,15–17]. Furthermore, longitudinal data from PTs have been used to assess the impact of the COVID-19 pandemic on academic performance, although findings in this area have been contradictory [18,19].

In Peru, there is no PT developed by a national association of medical schools that systematically promotes its implementation across medical training institutions. As

a result, there are no standardized data available to objectively compare medical schools or to evaluate students' knowledge progression during and at the end of their training, regardless of each university's curricular design. To date, PTs in Peru have been implemented as isolated institutional initiatives [20,21]. As such, longitudinal analysis of student cohorts and the identification of factors associated with academic progression remain key areas for investigation. In this context, our primary objective was to analyze trends in progress test scores among students at a medical school in Peru.

## Materials and methods

### Study design and setting

We conducted a longitudinal study using secondary data from the School of Medicine at the University of Piura, located in Lima, Peru. This medical school has implemented PTs since its inception in 2017. We analyzed data collected between 2017 and 2024.

The PT used in this study is called "Annual Case-Based Medical Examination." It is administered once a year during the second week of December, at the end of the academic year. The design, management, and administration of the test are entirely performed by the School of Medicine itself. Student participation is voluntary. The test consists of a written examination with 250 multiple-choice questions, each with a single correct answer. From 2017 to 2021, each question included four distractors and one correct option; starting in 2022, three distractors were used. The PT is organized into three booklets, each containing questions based on four to six narrative clinical cases. These cases are drawn from case reports published in peer-reviewed medical journals. The case descriptions, including relevant images and tables, are presented in full. The formulation of questions derived from these cases is supervised by course coordinators.

A distinctive feature of the progress test at this university is that it evaluates students based on their current stage of training, rather than on the level of knowledge expected at the completion of the medical program. Accordingly, the test content is limited to the subjects completed by the end of the academic year in which the assessment is administered. For instance, a third-year student is assessed on content from subject completed during the first, second, and third years. The number of questions per subject is proportional to the number of academic credits assigned to each course [21] (Fig 1).

The PT was administered virtually in 2020 and 2021 due to social restrictions imposed by the COVID-19 pandemic. In all other years, the test was conducted in person on the university campus using printed questionnaires.

### Participants

For the longitudinal analysis, we included students who completed six years of study prior to the medical internship (entry cohorts of 2017, 2018, and 2019) and participated in three or more PTs between 2017 and 2024. Students who discontinued their studies during the observation period were excluded from the analysis.

A multiple-group cross-sectional analysis was conducted to compare progress test scores between students in the preclinical phase (first to third year, focused on basic sciences) and those in the clinical phase (fourth to sixth year, focused on clinical sciences). This analysis included students who completed progress tests administered between 2019 and 2022. In both analyses, all students who met the inclusion criteria were included.

### Variables

The primary variable of interest was the PT score, which ranges from 0 to 250 points, with one point awarded for each correct answer. The raw score was subsequently converted to a 20-point scale, where a score of 10 corresponds to answering 50% of the questions correctly. Incorrect answers did not reduce points. The number of PT taken was considered a discrete variable, with values ranging from one to seven. As participation in the test is voluntary, not all students completed the six potential tests; some students took up to seven tests due to repeating academic years. Additional variables included the year of entry (cohorts from 2017 to 2024), the year of test administration (2017–2024), and sex (male/female). All data were provided by the Assessment Unit of the School of Medicine on 03/02/2025.

  

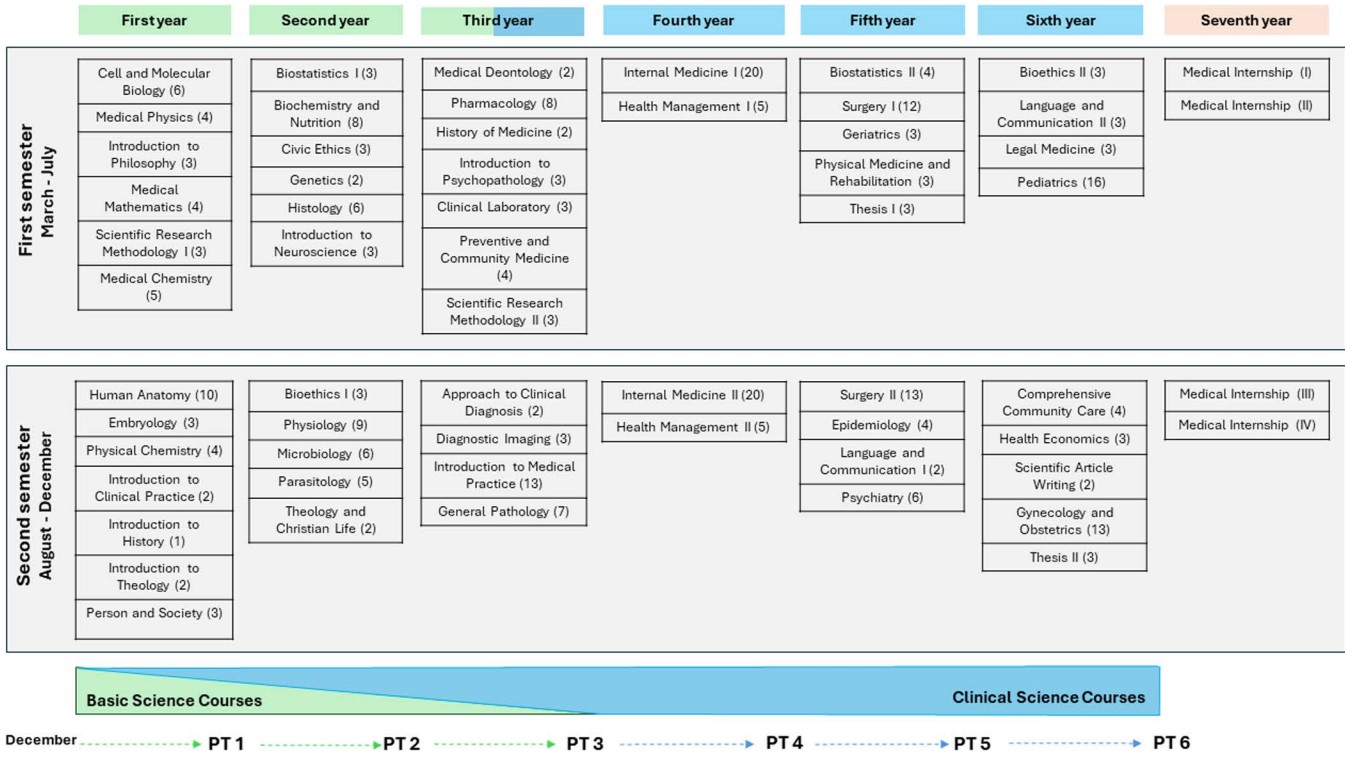

**Fig 1. Curriculum of the School of Medicine at the University of Piura, showing the courses and their corresponding academic credits (in parentheses) in relation to the annual administration of the progress test (PT).** PT1 refers to the progress test administered at the end of the first year, and so on.

## Data analysis

A descriptive analysis was conducted to examine participants by sex, entry cohort, and the number of PTs taken, using both absolute and relative frequencies. The median and interquartile range of the number of PTs were calculated for each sex and entry cohort. PTs scores across all years were summarized using the mean and standard deviation, while the distribution of scores was presented using a histogram. The histogram intervals were set to a width of 1. Furthermore, the distribution of scores for each year of test administration was illustrated using dot plots. Pearson correlation coefficients were calculated to assess the relationships between PTs scores across different years. The strength of the correlations was interpreted as follows: 0 to 0.10 (negligible), 0.11 to 0.39 (weak), 0.40 to 0.69 (moderate), 0.70 to 0.89 (strong), and 0.90 to 1.00 (very strong) [22].

The comparison of PTs scores between entry cohorts and years of test administration was performed using non-parametric tests: the Mann-Whitney U test for comparisons between two groups, and the Kruskal-Wallis test for comparisons among three or more groups. Post-hoc multiple comparisons were conducted using the Dunn test.

To assess the trend in PTs scores, we applied a generalized estimating equations (GEE) model with an identity link function, assuming a Gaussian distribution for the outcome. The standard error was adjusted for clustering by a unique numeric identifier, with an exchangeable correlation structure. The model included the year of test administration and was adjusted for the number of PTs taken and sex. Subsequently, using the *margins* command, we estimated the predictive margins for PTs scores for each year of administration, accounting for the estimated coefficients and adjusted standard errors from the GEE model. These estimates were calculated separately by entry cohort and the number of PTs taken.

We used the same generalized estimating equations (GEE) model to assess the trend in scores within the 2017, 2018, and 2019 entry cohorts, including the year of test administration and the number of PTs taken as variables. The GEE analysis was performed in STATA version 16 using the *xtgee* command. Graphs were generated using GraphPad Prism 10.4.2. A statistical significance level of 5% was considered.

## Ethical considerations

The study protocol was exempted from review by the Institutional Ethics Committee of the University of Piura, as the data were collected during routine academic activities and the analyzed database was fully anonymized. Prior to this exemption, the committee verified that obtaining informed consent was not feasible due to the nature of the secondary database analysis.

## Results

A total of 1,899 progress test scores from 669 medical students were included in the analysis. Of these, 369 (55.2%) were female. The first cohort (2017) included 51 students, while the most recent cohorts (2023 and 2024) each comprised 124 students. Among the participants, 7.6% (n = 51) did not take any PTs. Furthermore, 27.1% (n = 181) of students completed between four and five PTs from 2017 to 2024. Older entry cohorts tended to have a higher median number of progress tests completed (Table 1).

### Overall analysis of scores

The mean of the 1,899 PTs scores was 9.19, with a standard deviation of 2.34 (Fig 2A). The median score was 9.20, with an interquartile range (IQR) of 7.72 to 10.72. The minimum score recorded was 0.56, and the maximum was 16.72. In the PT administered in 2018, the median score was 11 (IQR = 9.42–12.16), while the lowest median score was observed in 2023, at 8.10 (IQR = 6.84–9.44). The PT administered in 2020 and 2021 were conducted virtually, with median scores of 10.24 and 10.48, respectively (Fig 2B).

PT scores from a given year demonstrated moderate to strong positive linear correlations with scores from the subsequent five years of test administrations. However, the 2017 PT scores did not show a significant correlation with those from the 2023 and 2024 administrations. Similarly, the 2018 PT scores were not significantly correlated with those from 2024 (Fig 3).

In the PT administered in 2019, no significant differences were observed in the median scores between the entry cohorts in 2017–2019 (H = 1.785, p = 0.41). In the PT of 2020, the 2019 entry cohort undertook clinical science subjects; however, no differences were found in their scores compared to the cohorts taking basic science subjects. In contrast, in the 2021 and 2022 PTs, students who transitioned to clinical subjects had significantly higher median scores than those in basic science subjects (Table 2).

### Longitudinal analysis of scores

A total of 167 students were part of the 2017, 2018, and 2019 entry cohorts, of whom 12 completed three or fewer tests. The longitudinal analysis included data from 155 students, 81 (52.3%) of whom were male. By entry cohort, 45 (29.0%) were from 2017, 55 (35.5%) from 2018, and 55 (35.5%) from 2019. Regarding the number of tests taken, 18 (11.6%) students completed four tests, 66 (42.6%) completed five, 61 (39.4%) completed six, and 10 (6.5%) completed seven tests.

In 2018, 2020, and 2021, the median scores were above 10, while in the other years, the median was below half of the 20-point scale. In 2018, 2020, and 2017, the highest scores exceeded 15 (16.72, 16.0, and 15.28, respectively). Regular students may voluntarily take up to six PTs during their medical training. However, PT scores from students in the 2017 cohort were still observed in the 2023 and 2024 administrations, and from the 2018 cohort in the 2024 PT (Fig 4A). The

**Table 1. Characteristics of medical students and the number of progress tests taken from 2017 to 2024.**

| Variable | Frequency | Percentage | Number of progress test | | |
|---|---|---|---|---|---|
| | | | Median | P25 | P75 |
| Sex | | | | | |
| Male | 300 | 44.8 | 3 | 1 | 5 |
| Female | 369 | 55.2 | 2 | 1 | 4 |
| Entry cohort | | | | | |
| 2017 | 51 | 7.6 | 5 | 4 | 6 |
| 2018 | 57 | 8.5 | 6 | 5 | 6 |
| 2019 | 59 | 8.8 | 5 | 5 | 6 |
| 2020 | 67 | 10 | 5 | 4 | 5 |
| 2021 | 96 | 14.3 | 3 | 1 | 4 |
| 2022 | 91 | 13.6 | 3 | 2 | 3 |
| 2023 | 124 | 18.5 | 2 | 1 | 2 |
| 2024 | 124 | 18.5 | 1 | 1 | 1 |
| Number of progress test | | | | | |
| None | 51 | 7.6 | --- | --- | --- |
| One | 173 | 25.9 | --- | --- | --- |
| Two | 126 | 18.8 | --- | --- | --- |
| Three | 67 | 10 | --- | --- | --- |
| Four | 68 | 10.2 | --- | --- | --- |
| Five | 113 | 16.9 | --- | --- | --- |
| Six | 61 | 9.1 | --- | --- | --- |
| Seven | 10 | 1.5 | --- | --- | --- |

P25: 25th percentile, P75: 75th percentile.

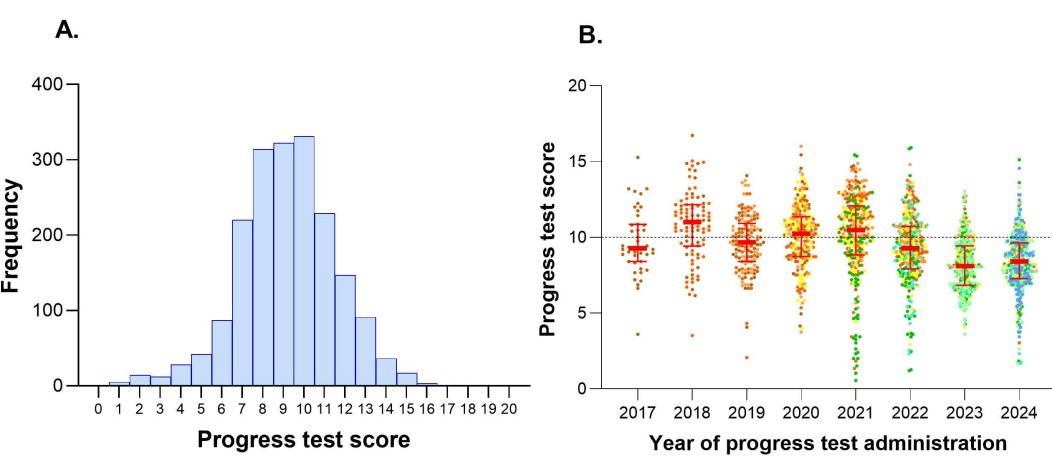

**Fig 2. (A) Histogram of the 1,899 progress test scores taken between 2017 and 2024. (B) Scatter plot of progress test scores by year of administration.** The colors of the points reflect the scores of students from a particular entry cohort. In the 2017 plot, only one color is used, in the 2018 plot, two colors (representing two cohorts), and so on. The red central lines represent the median, and the bars represent the 1st and 3rd quartiles. The dashed black horizontal line indicates half of the score on the 20-point scale.

| Year of administration | | 2017 | 2018 | 2019 | 2020 | 2021 | 2022 | 2023 | 2024 |
|---|---|---|---|---|---|---|---|---|---|
| **2017** | r | 1.000 | 0.585 | 0.708 | 0.502 | 0.495 | 0.711 | 0.464 | 0.500 |
| | p | - | <0.001 | <0.001 | 0.001 | 0.001 | <0.001 | 0.294 | 0.667 |
| | n | 44 | 34 | 38 | 39 | 42 | 22 | 7 | 3 |
| **2018** | r | 0.585 | 1.000 | 0.802 | 0.551 | 0.482 | 0.643 | 0.692 | -0.032 |
| | p | <0.001 | - | <0.001 | <0.001 | <0.001 | <0.001 | <0.001 | 0.905 |
| | n | 34 | 94 | 83 | 89 | 90 | 68 | 42 | 16 |
| **2019** | r | 0.708 | 0.802 | 1.000 | 0.598 | 0.596 | 0.645 | 0.73 | 0.431 |
| | p | <0.001 | <0.001 | - | <0.001 | <0.001 | <0.001 | <0.001 | 0.008 |
| | n | 38 | 83 | 141 | 135 | 134 | 109 | 83 | 37 |
| **2020** | r | 0.502 | 0.551 | 0.598 | 1.000 | 0.657 | 0.583 | 0.55 | 0.589 |
| | p | 0.001 | <0.001 | <0.001 | - | <0.001 | <0.001 | <0.001 | <0.001 |
| | n | 39 | 89 | 135 | 220 | 207 | 175 | 147 | 97 |
| **2021** | r | 0.495 | 0.482 | 0.596 | 0.657 | 1.000 | 0.668 | 0.588 | 0.542 |
| | p | 0.001 | <0.001 | <0.001 | <0.001 | - | <0.001 | <0.001 | <0.001 |
| | n | 42 | 90 | 134 | 207 | 293 | 232 | 196 | 144 |
| **2022** | r | 0.711 | 0.643 | 0.645 | 0.583 | 0.668 | 1.000 | 0.727 | 0.678 |
| | p | <0.001 | <0.001 | <0.001 | <0.001 | <0.001 | - | <0.001 | <0.001 |
| | n | 22 | 68 | 109 | 175 | 232 | 306 | 244 | 189 |
| **2023** | r | 0.464 | 0.692 | 0.73 | 0.55 | 0.588 | 0.727 | 1.000 | 0.631 |
| | p | 0.294 | <0.001 | <0.001 | <0.001 | <0.001 | <0.001 | - | <0.001 |
| | n | 7 | 42 | 83 | 147 | 196 | 244 | 386 | 288 |
| **2024** | r | 0.500 | -0.032 | 0.431 | 0.589 | 0.542 | 0.678 | 0.631 | 1.000 |
| | p | 0.667 | 0.905 | 0.008 | <0.001 | <0.001 | <0.001 | <0.001 | - |
| | n | 3 | 16 | 37 | 97 | 144 | 189 | 288 | 415 |

**Fig 3. Heat map of the Pearson correlation coefficients between progress test scores across test administrations from 2017 to 2024.** Light blue: 0.40 a 0.69 (moderate correlation), dark blue: 0.70 a 0.89 (strong correlation). P values > 0.05 were considered non-significant correlations. *n* is the number of valid data pairs used to assess the correlation.

average scores of the PTs were higher for students who completed five tests compared to those who completed four or seven tests (Fig 4B).

No clear trend was observed in the distribution of PTs scores across any of the entry cohorts. In the 2017 cohort, significant differences were found in the median scores (H statistic = 31.5, p < 0.001), primarily due to the higher median scores in 2021 (Fig 5A). In the 2018 cohort, significant differences were also identified in the median scores (H statistic = 20.2, p = 0.001), attributed to the lower median scores in 2023 compared to 2021 and 2018 (Fig 5B). Finally, in the 2019 cohort, the difference in medians (H statistic = 44.5, p < 0.001) was explained by the lower median scores in the last two years compared to 2020, 2021, and 2022 (Fig 5C).

**Table 2. Comparison of progress test scores from 2019 to 2022 among cohorts that completed the first three years of study, based on entry cohort and transition to clinical subjects.**

| Year of progress test administration | Entry cohort | n | P50 | P25 | P75 | Statistic | P value |
|---|---|---|---|---|---|---|---|
| 2019 | 2017 | 42 | 9.44 | 8.44 | 10.48 | 1.785 [a] | 0.410 |
| | 2018 | 51 | 9.84 | 8.4 | 11.28 | | |
| | 2019 | 48 | 9.64 | 8.5 | 11.16 | | |
| 2020 | 2017 | 46 | 10.24 | 8.88 | 11.96 | 5.464 [a] | 0.141 |
| | 2018 | 54 | 9.4 | 8.1 | 11.24 | | |
| | 2019 | 56 | 10.56 | 9.4 | 11.62 | | |
| | 2020 | 64 | 10.48 | 8.74 | 11.32 | | |
| 2021 | 2017 | 45 | 11.44 | 10.2 | 12.8 | 41.32 [a] | <0.001 |
| | 2018 | 56 | 10.88 | 8.78 | 12.62 | | |
| | 2019 | 56 | 11.2 | 9.4 | 13.18 | | |
| | 2020 | 61 | 10.4 | 9.44 | 11.52 | | |
| | 2021 | 75 | 9.04 | 5.76 | 10.72 | | |
| 2022 | 2017 | 25 | 9.76 | 8.32 | 11.2 | 33.71 [a] | <0.001 |
| | 2018 | 50 | 10.16 | 9.02 | 11.86 | | |
| | 2019 | 51 | 9.6 | 8.56 | 10.72 | | |
| | 2020 | 56 | 9.28 | 7.76 | 10.56 | | |
| | 2021 | 57 | 7.84 | 6.72 | 9.56 | | |
| | 2022 | 67 | 9.2 | 7.6 | 10.8 | | |
| | **Transition to clinical subjects** | | | | | | |
| 2020 | Yes | 46 | 10.24 | 8.88 | 11.96 | 0.314 [b] | 0.575 |
| | No | 174 | 10.24 | 8.72 | 11.36 | | |
| 2021 | Yes | 101 | 11.36 | 9.56 | 12.68 | 11.621 [b] | 0.001 |
| | No | 192 | 10.08 | 8.36 | 11.68 | | |
| 2022 | Yes | 126 | 9.84 | 8.78 | 11.08 | 22.05 [b] | <0.001 |
| | No | 180 | 8.88 | 7.36 | 10.4 | | |

[a] H=H – Kruskal-Wallis,

[b] U – Mann-Whitney. P50=Median, P25: 25th percentile, P75: 75th percentile.

In the generalized estimating equations (GEE) model, no consistent upward trend in scores over time was observed. When we formulated a model with time as a numerical variable, we found that, on average, scores decreased by 0.088 points per additional year (SE = 0.030, 95% CI: −0.147 to −0.029, p = 0.003), after controlling for the number of PT taken.

Another model, presented in Table 3, included time (year of PT administration), the number of PTs, and sex as explanatory variables. In the full model, the years 2021 and 2018 showed significant increases in scores compared to 2017 (+1.38 and +1.21, respectively), while scores in 2019, 2023, and 2024 did not differ significantly from those in 2017. These results remained consistent when the model only considered the number of PTs taken. In the reduced model, regardless of time, taking five PTs resulted in a 1.40-point increase in scores compared to taking four PTs, while taking six tests resulted in a 0.87-point increase compared to four. No clear trend of increasing scores with a higher number of PTs taken was observed. When the predicted scores, based on the GEE model estimates, were graphed, it was evident that, in all years of PT administration, students who took five PTs had higher scores (Fig 6A). Similarly, when scores were analyzed by entry cohort, all cohorts showed a sustained decline in scores from 2022 to 2024 (Fig 6B).

When replicating the analysis for each entry cohort, different trends in scores over time were observed. In the 2017 cohort, a significant increasing trend in scores was found between 2018 and 2021, while no significant differences were observed in 2023 and 2024 compared to 2017. In the 2018 cohort, scores in 2019 and 2020 decreased significantly compared to the

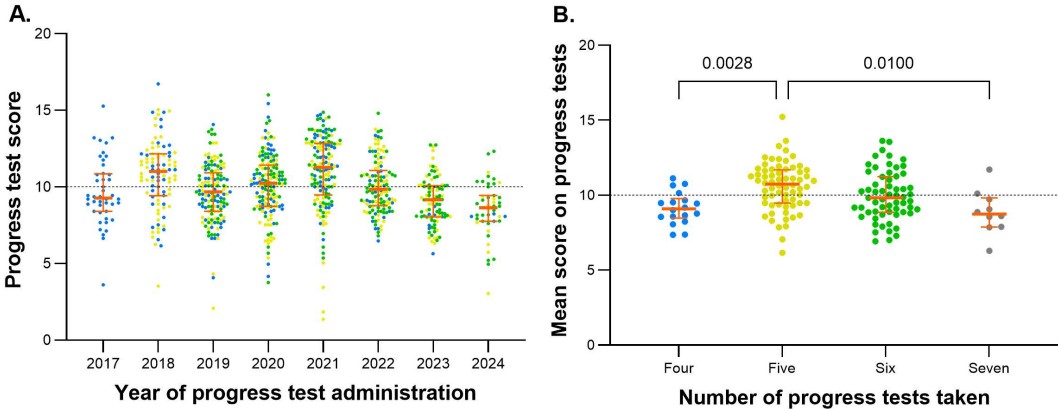

**Fig 4. (A) Scatter plot of progress test scores by year of administration.** The entry cohort of 2017 is represented by blue points, the 2018 entry cohort by yellow points, and the 2019 entry cohort by green points. **(B) Scatter plot of average progress test scores grouped by the number of tests taken.** The bars above the points represent p-values from Dunn's multiple comparisons tests. The dashed black horizontal line indicates half of the maximum score on a 20-point scale. The red central lines correspond to the median, and the bars represent the 1st and 3rd quartiles.

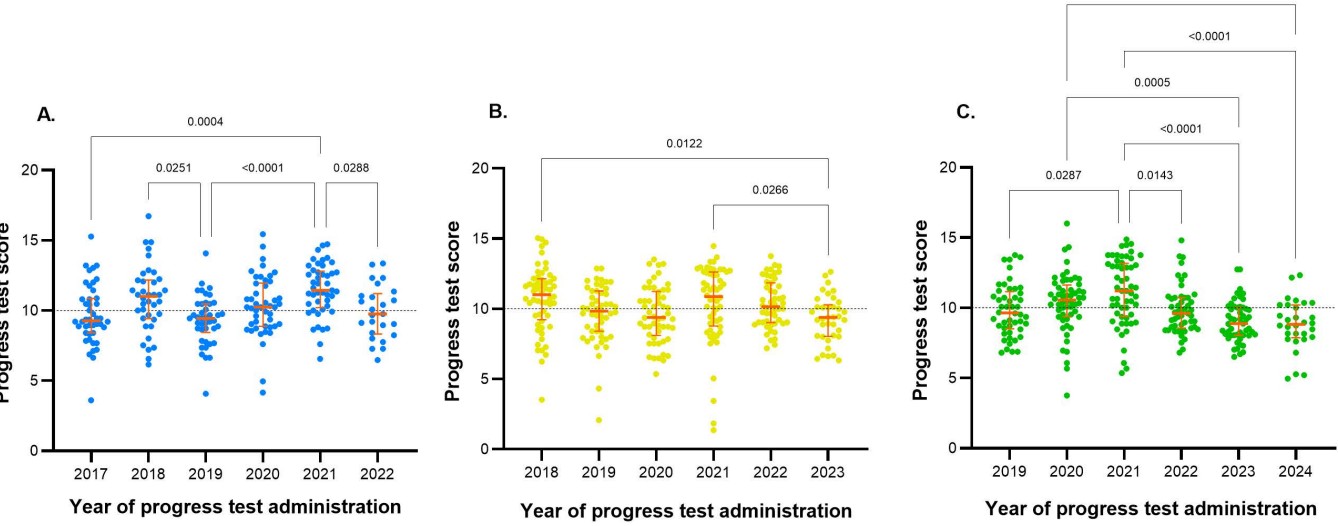

**Fig 5. Scatter plot of scores in the first six progress tests by entry cohort.** (A) 2017 entry cohort, (B) 2018 entry cohort, (C) 2019 entry cohort. All cohorts had the opportunity to complete at least six progress tests before the medical internship. The dashed black horizontal line indicates half of the maximum score on a 20-point scale. The red central lines correspond to the median, and the bars represent the 1st and 3rd quartiles. The upper bars indicate p-values obtained from Dunn's multiple comparisons test.

first PT. Subsequently, between 2022 and 2024, a declining trend in scores was observed. In the 2019 cohort, a significant increase in scores was seen in 2020 and 2021 compared to the baseline (2019), followed by a decrease in scores in 2022 and 2023, with the reduction in 2023 being significant. In all cohorts, taking five PTs was associated with higher scores (Table 4).

## Discussion

We did not observe a clear and sustained upward trend in the scores across eight PTs administered at a medical school in Peru. Although the score trajectories varied by entry cohort, the scores, on average, showed a stable pattern. Students



**Table 3. Generalized Estimating Equations (GEE) models to assess the evolution of scores based on time, number of progress tests, and sex.**

| Variable | Model 1 | | | | | Model 2 | | | | |
|---|---|---|---|---|---|---|---|---|---|---|
| | β | SE | CI95% | | P value | β | SE | CI95% | | P value |
| | | | LL | UL | | | | LL | UL | |
| Intersection | 8.854 | 0.317 | 8.23 | 9.48 | <0.001 | 8.656 | 0.301 | 8.066 | 9.246 | <0.001 |
| Time | | | | | | | | | | |
| 2017 | Ref. | | | | | Ref. | | | | |
| 2018 | 1.216 | 0.274 | 0.678 | 1.755 | <0.001 | 1.216 | 0.275 | 0.678 | 1.755 | <0.001 |
| 2019 | −0.024 | 0.244 | −0.503 | 0.454 | 0.921 | −0.025 | 0.244 | −0.504 | 0.454 | 0.917 |
| 2020 | 0.514 | 0.268 | −0.011 | 1.038 | 0.055 | 0.511 | 0.268 | −0.013 | 1.036 | 0.056 |
| 2021 | 1.381 | 0.274 | 0.844 | 1.917 | <0.001 | 1.379 | 0.274 | 0.842 | 1.915 | <0.001 |
| 2022 | 0.481 | 0.254 | −0.018 | 0.980 | 0.059 | 0.478 | 0.254 | −0.020 | 0.977 | 0.06 |
| 2023 | −0.355 | 0.276 | −0.895 | 0.185 | 0.198 | −0.360 | 0.275 | −0.899 | 0.180 | 0.191 |
| 2024 | −0.437 | 0.350 | −1.123 | 0.250 | 0.212 | −0.440 | 0.350 | −1.126 | 0.247 | 0.209 |
| Number of progress test | | | | | | | | | | |
| Four | Ref. | | | | | Ref. | | | | |
| Five | 1.342 | 0.309 | 0.736 | 1.948 | <0.001 | 1.403 | 0.309 | 0.797 | 2.009 | <0.001 |
| Six | 0.847 | 0.319 | 0.221 | 1.473 | 0.008 | 0.875 | 0.320 | 0.248 | 1.503 | 0.006 |
| Seven | −0.295 | 0.494 | −1.263 | 0.674 | 0.628 | −0.193 | 0.504 | −1.181 | 0.795 | 0.702 |
| Sex | | | | | | --- | | | | |
| Male | Ref. | | | | | --- | | | | |
| Female | −0.327 | 0.243 | −0.803 | 0.149 | 0.179 | --- | | | | |

GEE model with an identity link function, Gaussian family, and exchangeable correlation structure. Standard errors adjusted for participant ID clustering.

Ref. = reference, SE = standard error, CI = confidence interval (lower limit: LI, upper limit: LS).

in the 2017 cohort had some years in which scores were higher than those obtained in the entry year. In the 2018 cohort, scores decreased in 2019, 2020, and 2024 compared to the first PT. In the 2019 cohort, score increases were observed only in 2020 and 2021, compared to the first year. In all PTs, the median scores tended to remain around half of the maximum score on the 20-point scale.

Our data show a moderate to strong linear correlation among the scores of the first five PTs. However, this correlation disappears for the sixth and seventh tests. Several factor may explain this finding. Across all entry cohorts and years of test administration, students who completed five PTs had higher scores compared to those who completed four or seven PTs. This pattern can be explained by the characteristics of our PT and the curriculum progression. The maximum number of tests a student can complete, assuming they pass all courses and fulfill the required academic credits each semester, is six. Students who completed seven PTs are those who were not promoted to the next year at some point in the curriculum, while those who completed four or fewer tests likely had insufficient and inconsistent exposure to the PT, mainly due to its non-mandatory nature at this medical school. Another possible explanation for declining participation over time is a decrease in student´s favorable perceptions of the PTs after repeated administrations. For example, students who completed five PTs reported lower satisfaction levels compared to those who had taken only one [23]. Additionally, given the non-mandatory nature of this PT experience, students in their final years may have opted out to focus on targeted preparation for the national medical licensing examination.

In our study, using a 20-point scale, the average scores did not exceed half of the maximum score. Additionally, in the longitudinal analysis, we consistently observed that scores did not increase as students progressed through the curriculum. This contrasts with previous studies where PTs were designed to measure cumulative knowledge expected at the end of medical training—often showing a clear increase in scores over time. In our context, however, the PT specifically

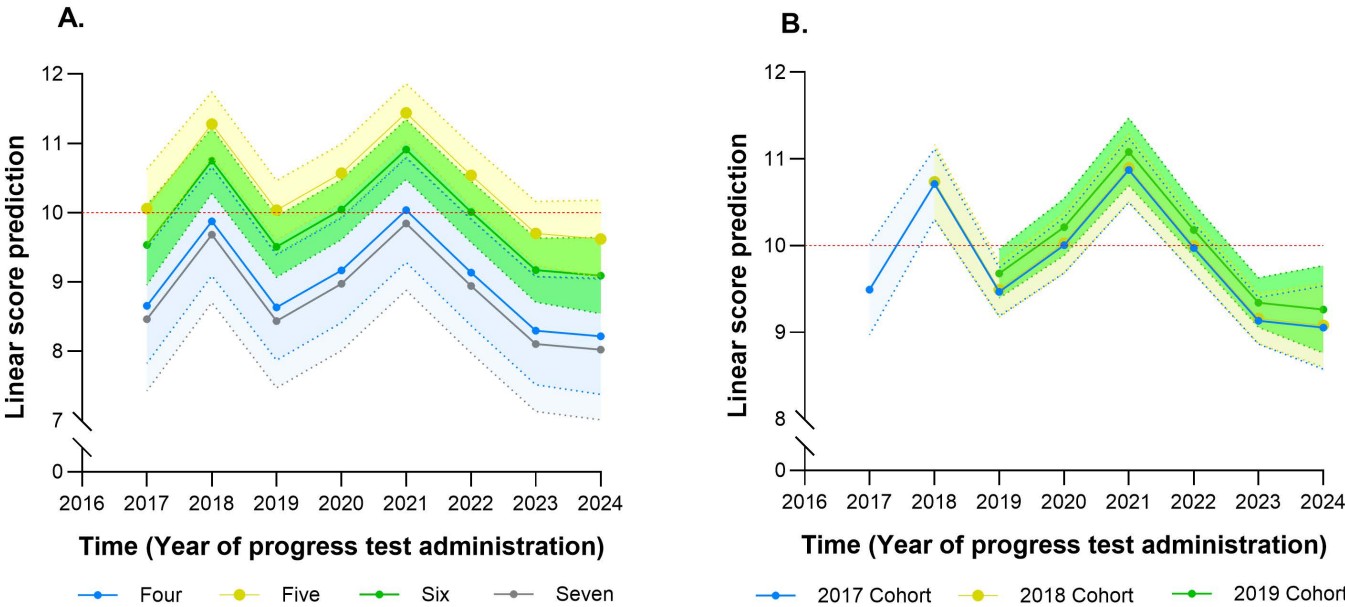

**Fig 6. Prediction of progress test scores based on the number of tests taken (A) and entry cohort (B).** The predictions were based on estimates from generalized estimating equations (GEE) models with the following independent variables: year of progress test administration, number of progress tests taken, and sex.

assessed students' knowledge based on the curricular content covered up to the point of each test administration. This design feature appears to be a key factor underlying the observed score stability across academic years. Therefore, rather than indicating a lack of academic progress, the absence of score growth in our study likely reflects the formative nature of the test and its alignment with students' stage-specific learning milestones.

As previously mentioned, longitudinal studies using traditional PTs have reported score increases of varying magnitudes. A study conducted at the medical schools of the Universities of Groningen, Maastricht, and Nijmegen observed an increase in the average proportion of correct answers from 5.7% to 67.5% between the first and last PT, by the end of the sixth year of study [24]. At McMaster University, a similar trend was observed in four entry cohorts, with the proportion of correct answers rising from approximately 12–18% on the first test to nearly 50% on the final test [4]. In Saudi Arabia, an analysis of progress test results collected over a 10-year period from multiple medical schools found that the proportion of correct answers increased from 6% in the first year to 38.2% in the fifth year [17]. This pattern was also observed at the University of São Paulo in Brazil, where the proportion of correct responses rose from 32% in the first year to 56.5% in the sixth year [25]. This upward trajectory was further corroborated by a cross-sectional analysis of a 2015 national PT administered in several Brazilian medical schools, which showed that first-year students achieved an average of 32.38% correct responses, compared to 61.28% among sixth-year students [7].

Although our findings on PTs score trends are not directly comparable to those reported in most previous studies, we observed that medical students, on average, achieved scores equivalent to approximately 50% of the maximum possible on each administration of the test. This level of correct responses is comparable to that reported in other settings for students taking their final PT, typically at the end of the curriculum, after completion of the full curriculum [4,15,24,25]. Despite these results, 73.4% of students exposed to this PT reported that it helped them apply their knowledge to clinical contexts, 60.8% felt it allowed them to demonstrate their knowledge, and 52.1% considered it a fair assessment [23]. Additionally, among 26 graduates from the 2017 cohort and 25 from the 2018 cohort who took the Peruvian national medical licensing examination in 2023 and 2024, respectively, none failed the exam. These facts suggest that the

**Table 4. Evolution of progress test scores in medical students by entry cohort.**

| Variable | Cohort 2017 | | | | | Cohort 2018 | | | | | Cohort 2019 | | | | |
|---|---|---|---|---|---|---|---|---|---|---|---|---|---|---|---|
| | β | SE | CI95% | | p | β | SE | CI95% | | p | β | SE | CI95% | | p |
| | | | LL | UL | | | | LL | UL | | | | LL | UL | |
| Intersection | 8.75 | 0.31 | 8.14 | 9.35 | <0.001 | 9.13 | 0.78 | 7.60 | 10.67 | <0.001 | 8.79 | 0.42 | 7.96 | 9.62 | <0.001 |
| Time | | | | | | | | | | | | | | | |
| 2017 | Ref. | | | | | --- | --- | --- | ---- | --- | --- | --- | --- | --- | --- |
| 2018 | 0.99 | 0.30 | 0.40 | 1.58 | 0.001 | Ref. | | | | | --- | --- | --- | --- | --- |
| 2019 | −0.44 | 0.22 | −0.88 | 0.00 | 0.050 | −1.15 | 0.23 | −1.60 | −0.70 | <0.001 | Ref. | | | | |
| 2020 | 0.71 | 0.32 | 0.07 | 1.34 | 0.030 | −1.26 | 0.26 | −1.78 | −0.75 | <0.001 | 0.70 | 0.25 | 0.21 | 1.20 | 0.005 |
| 2021 | 1.78 | 0.29 | 1.20 | 2.35 | <0.001 | −0.52 | 0.42 | −1.34 | 0.31 | 0.223 | 1.55 | 0.24 | 1.09 | 2.01 | <0.001 |
| 2022 | 0.66 | 0.28 | 0.10 | 1.21 | 0.020 | −0.61 | 0.28 | −1.16 | −0.05 | 0.032 | 0.09 | 0.21 | −0.31 | 0.50 | 0.656 |
| 2023 | −0.19 | 0.57 | −1.31 | 0.94 | 0.742 | −1.31 | 0.26 | −1.82 | −0.80 | <0.001 | −0.72 | 0.20 | −1.12 | −0.33 | <0.001 |
| 2024 | 0.55 | 0.48 | −0.39 | 1.49 | 0.252 | −1.93 | 0.71 | −3.33 | −0.54 | 0.007 | −0.53 | 0.29 | −1.10 | 0.05 | 0.071 |
| Number of progress test | | | | | | | | | | | | | | | |
| Four | Ref. | | | | | Ref. | | | | | Ref. | | | | |
| Five | 1.95 | 0.48 | 1.01 | 2.89 | <0.001 | 2.10 | 0.82 | 0.50 | 3.70 | 0.010 | 1.12 | 0.48 | 0.17 | 2.07 | 0.021 |
| Six | 0.73 | 0.50 | -0.25 | 1.70 | 0.145 | 1.69 | 0.81 | 0.10 | 3.27 | 0.037 | 0.91 | 0.55 | -0.16 | 1.98 | 0.097 |
| Seven | −0.75 | 0.41 | −1.54 | 0.05 | 0.066 | 0.86 | 0.99 | −1.08 | 2.80 | 0.386 | --- | --- | --- | --- | --- |

GEE model with an identity link function, Gaussian family, and exchangeable correlation structure. Standard errors adjusted for participant ID clustering.

Ref. = reference, SE = standard error, CI = confidence interval (lower limit: LI, upper limit: LS).

Cohort 2017: Number of observations = 238, number of participants = 45.

Cohort 2018: Number of observations = 312, number of participants = 55.

Cohort 2019: Number of observations = 288, number of participants = 55.

observed PT performance was expected and did not negatively affect students' academic progress or preparedness for licensure.

The analysis period included the COVID-19 pandemic. Despite this, the administration of PTs continued, although the testing modality changed in 2020 and 2021, shifting from in-person to online formats. During these two years, all three student cohorts obtained higher scores compared to the 2019 PT. This transition from face-to-face to virtual testing was adopted in several countries [18,19]. However, the impact of pandemic-related adaptations in medical education on performance in PTs remains a subject of ongoing debate. For instance, at the Charité – Universitätsmedizin Berlin, an increase in student scores was reported in the PTs administered in April and November 2022, compared to previous exams [19]. In contrast, a study conducted at two universities in Brazil found that students enrolled in clinical subjects, who experienced suspended hospital rotations during the pandemic, did not show a significant increase in scores in the 2020 PT compared to 2019 [18].

In 2022, PTs returned to an in-person format using printed materials. That year, a significant change was implemented: the number of distractors per item reduced from four to three. Among students from the 2017–2019 entry cohorts, we observed a sustained decline in PTs scores from 2022 onward. Reducing the number of distractors can help eliminate non-functional distractors, thereby increasing the distractor efficiency of the remaining items. This adjustment is directly related to the item difficulty index [26]. Studies have shown that in multiple-choice questions with three distractors, when all distractors are functional, the average item difficulty is around 56%. However, when only one distractor is functional, the item difficulty increases to approximately 74% [27]. An item difficulty index of 80% or higher typically indicates that a question is easier to answer [28].

Another key finding was the increase in PTs scores when students began their clinical subjects in the fourth year of study. No significant differences in scores were observed during the first three years, which focus on basic sciences. This result aligns with a previous study conducted in the same institution, using the same PT. In that study, students who had already started their clinical training scored higher on questions related to content from the first two years, even outperforming students currently taking those early-year subjects [29]. This improvement may be attributed to the enhanced vertical integration activities during the clinical training phase, which likely reinforce and consolidate the knowledge gained in earlier stages of the curriculum [30].

The study has limitations that should be considered when interpreting the results. The PT analyzed assesses knowledge related to the subjects completed by students at the time of the test. Most international experiences with PTs evaluate the expected knowledge at graduation, which complicates the comparison of our findings. However, it provides new insights into this particular type of PT. With the data analyzed, we were unable to conduct a psychometric evaluation of the eight PTs. This information would have provided greater context to explain the longitudinal trends in scores. This analysis pertains to a single medical school, so the findings and longitudinal score trajectories cannot be generalized to other public or private universities in Peru. However, it demonstrates the feasibility of implementation in the Peruvian context. Two characteristics of the PT should be considered. First, the non-mandatory nature of the test meant that not all students had the opportunity to take all six tests, which affected the estimation of longitudinal trajectories. Additionally, some irregular students, who failed courses and were not promoted to the next year, took more PTs than expected. On average, these students performed worse in these evaluations.

Based on these findings, we propose several recommendations. The developers of the analyzed PT could include a subset of questions that assess the knowledge expected of a medical student upon graduation. This would provide additional pedagogical insights from this PT experience. The analysis of longitudinal performance suggests that students should take a minimum of five PTs. The transition from basic sciences to clinical subjects in the curriculum leads to improvements in PT performance. However, the observed increase in the median scores by one point could be further enhanced by strengthening vertical integration strategies in the curriculum. Finally, it is necessary to evaluate the reliability and validity of the applied PTs.

## Conclusion

This PT experience, designed to assess students' knowledge based on their progression through the curriculum, provides longitudinal data that do not show a clear and sustained upward trend in PTs scores. This stable pattern of performance was consistently observed across all analyzed entry cohorts. Although the median scores fluctuated within each cohort, they tended to stabilize around the midpoint of the 20-point scale. Students who completed five or six PTs achieved higher predicted scores across all years of test administration. However, this association may be influenced by overall academic performance, as students who took five PTs were typically regular students who did not fail subjects. Finally, the transition from basic science to clinical subjects in the curriculum coincided with an increase in median scores.

## Supporting information

**S1 Appendix. Raw database.**
(XLSX)

## Author contributions

**Conceptualization:** Franco Romani, Cesar Gutierrez.

**Data curation:** Franco Romani.

**Formal analysis:** Franco Romani.



**Investigation:** Franco Romani, Cesar Gutierrez.

**Methodology:** Franco Romani, Cesar Gutierrez.

**Supervision:** Franco Romani, Cesar Gutierrez.

**Writing – original draft:** Franco Romani.

**Writing – review & editing:** Franco Romani, Cesar Gutierrez.

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
