## [Decision Letter · Decision Letter 0]

23 Jun 2025

PONE-D-25-22024Trends in progress test performance of medical students at a university in PeruPLOS ONE

Dear Dr. Romani,

Thank you for submitting your manuscript to PLOS ONE. After careful consideration, we feel that it has merit but does not fully meet PLOS ONE’s publication criteria as it currently stands. Therefore, we invite you to submit a revised version of the manuscript that addresses the points raised during the review process.

**ACADEMIC EDITOR:**

Make point-by-point corrections to the issues highlighted by each reviewer.Carry out a thorough review of references cited.

We look forward to receiving your revised manuscript.

Kind regards,

Rosemary Bassey, Ph.D.

Academic Editor

PLOS ONE

Journal Requirements: 

 [Universidad de Piura (PI2501)]. 

Reviewers' comments:

Reviewer's Responses to Questions

**Comments to the Author**

1. Is the manuscript technically sound, and do the data support the conclusions?

Reviewer #1: Yes

Reviewer #2: Yes

2. Has the statistical analysis been performed appropriately and rigorously? 

Reviewer #1: Yes

Reviewer #2: Yes

3. Have the authors made all data underlying the findings in their manuscript fully available?

Reviewer #1: Yes

Reviewer #2: Yes

4. Is the manuscript presented in an intelligible fashion and written in standard English?

Reviewer #1: Yes

Reviewer #2: Yes

5. Review Comments to the Author

Reviewer #1: The authors use a technically sound approach to analyzing the data for a very interesting program utilizing progress testing. The following are concerns that should be addressed.

1. While the study is well done, the authors need to more carefully use the abstract to highlight the novel conclusions that are being drawn.

2. It is interesting to note that the authors of the exams have achieved a progressive level of difficulty that causes stable scores across the years of a student being in the medical school. A discussion of how this is achieved is surely warranted.

3. The difficulty of the exams (around 0.5) suggests the exams are very challenging. The authors should comment on whether this is desirable, and whether this low level of success is concerning in students that are about to graduate.

4. Were psychometric measures (e.g. difficulty, point biserial) used to evaluate test item performance? If so, were poor performing questions omitted, and is the analyzed data before or after omission?

5. Concerns are raised by references 3 and 22. These articles report recent studies that are by some of the the same authors using much of the same data. The authors should carefully review the conclusions of these earlier studies and what is novel in this the study. Reference 3 is used in the introduction instead of directly referencing early studies of progress testing. Is this choice necessary?

6. Line 62 should be "students", not "student".

Reviewer #2: The manuscript is well composed and delivers insights on an important assessment tool in medical education. The researchers have taken a sizeable sample and used reasonable statistics mostly through the research.

However, this reviewer would like to suggest some corrections or explanations to a few issues:

1. Language (translation) issues in cover page

2. Meaning of statement in line 34-35 not clear.

3. In line 120, the word "median score" is used in a non-statistical context. "Median" should strictly be used to describe the population and findings as one measure of central tendency.

4. Line 134: Use of Pearson correlation to compare test scores raises some concerns. If we are comparing test scores of the same student in a longitudinal study, paired tests or repeated ANOVA might be more justifiable.

5. Line 144 states that the test was somehow based on "participant identity" while line 158 says data was anonymized. It is possible to do per participant analysis in anonymized data. But the procedure should be mentioned in methodology. The statement raises ethical issues.

6. Declining interest in PTs throughout the years is very distinctly seen. Students are taking lesser and lesser tests as years pass. This should be discussed and reasons should be explored.

7. Mean and median both used to describe data in lines 172-173. It is better to use one measure of central tendency as per normality test of the variable.

8. In lines 186-189, are the years used to characterize cohort or year of test taken? As a reader, this reviewer couldn't identify it correctly. Same with Fig 3 caption. Are those years the cohort number? or the date of test?

9. Initiation of clinical subjects in table 2 is not clear

12. Line 215 states "required to complete", while elsewhere PTs are termed "voluntary". This confuses readers. Please try to use single narrative and explain this in methodology.

13. Taking the baseline score as covariate / confounder would be good during analysis so that students with better or worse baseline knowledge are accounted for. Correlation among scores of consecutive tests depends on baseline of each participant. Hence, it is better to account for it. Again, paired analysis could suit this situation better.

15. One big question that this reviewer pondered about was: Why is average PT around 50%? Why is it not progressing? Is it about the modality of PT? Is it about the medical education being delivered? The authors should try to explore these findings in detail.

6. PLOS authors have the option to publish the peer review history of their article (what does this mean? ). If published, this will include your full peer review and any attached files.

**Do you want your identity to be public for this peer review?** For information about this choice, including consent withdrawal, please see our Privacy Policy .

Reviewer #1: No

Reviewer #2: No

---

## [Author Response · Author response to Decision Letter 1]

25 Jun 2025

Dear editor,

We thank you for the time you spent reviewing our initial submission of our manuscript entitled “Trends in progress test performance of medical students at a university in Peru” (PONE-D-25-22024).

Below, we provide our point-by-point responses to each comment, indicating any corresponding changes made to the revised manuscript.

Reviewer #1: The authors use a technically sound approach to analyzing the data for a very interesting program utilizing progress testing. The following are concerns that should be addressed.

1. While the study is well done, the authors need to more carefully use the abstract to highlight the novel conclusions that are being drawn.

We appreciate the observation and agree that the conclusion could be more explanatory and persuasive considering the findings. Therefore, we have rewritten the conclusion section of the abstract to highlight the main and, in our view, most novel findings of our study, while also providing the most plausible explanation for these results. The new conclusion of the abstract is as follows:

Over an eight-year period of administering a progress test at a university in Peru, student performance remained stable, with an average of approximately 50% of questions answered correctly per test. Longitudinal analysis did not show a sustained increase in scores as students advanced through the curriculum. This pattern may be explained by the progress test design, which assesses only the content covered by students at the time of each administration, unlike other progress tests that measure end-of-curriculum knowledge across all cohorts. Nevertheless, an increase in median scores was observed during the transition from basic science to clinical subjects.

2. It is interesting to note that the authors of the exams have achieved a progressive level of difficulty that causes stable scores across the years of a student being in the medical school. A discussion of how this is achieved is surely warranted.

We appreciate the comment. Our analysis focused on evaluating longitudinal performance across eight progress tests administered annually. Determining the difficulty index of the applied progress tests, or assessing these indices across all eight tests, was not among our study objectives. Instead, our analysis aimed to examine the performance trend of students on these tests throughout their medical studies. The finding of a stable score, without a sustained upward trend and averaging around 50% of the maximum score, is addressed in the Discussion section (lines 307–315). To our understanding, this stable pattern is mainly explained by the characteristics of the progress test used. Specifically, it includes questions only from subjects that students have completed up to the time of each test. This differs from other international experiences, where progress tests include questions covering all subjects within the medical curriculum—that is, they assess the knowledge expected upon graduation. This explanation has also been included as a study limitation (lines 372–375). Furthermore, a recommendation related to this finding has been formulated: "The developers of the analyzed PT could include a subset of questions that assess the knowledge expected of a medical student upon graduation." (lines 386-388).

3. The difficulty of the exams (around 0.5) suggests the exams are very challenging. The authors should comment on whether this is desirable, and whether this low level of success is concerning in students that are about to graduate.

We appreciate this observation. According to the National Board of Medical Examiners (NBME), very easy items have a difficulty index >0.95, while very difficult items have an index <0.30. Such questions offer limited information about the population. Other studies analyzing Spain's medical residency entrance examination (MIR) consider difficulty indices between >0.5 and 0.6 to be excellent, and those between >0.3 and 0.5 to be acceptable.

Our study does not report item-level difficulty index for the progress tests, nor does it aim to provide summary measures of this indicator. However, our findings suggest that, on average, students answered approximately 50% of the questions correctly. The score distribution was approximately normal, with a minimum of 0.56 and a maximum of 16.72. As described in the Discussion (lines 324–329), this result is consistent with previous experiences using progress testing among students nearing the end of the medical curriculum.

Moreover, as acknowledged in the Discussion (lines 376–378), a limitation of our analysis is the lack of psychometric indicators for the progress tests evaluated, including item difficulty index.

Regarding concerns about the relatively modest performance of students close to graduation, a previous study conducted among students who took the same progress test reported that 73.4% felt the test helped them apply their knowledge to clinical contexts, 60.8% believed it allowed them to demonstrate their knowledge, and 52.1% considered it a fair assessment. Furthermore, among graduates from the 2017 (n=26) and 2018 (n=25) cohorts—who subsequently took the Peruvian national medical licensing examination in 2023 and 2024—none failed the exam. These results suggest that the outcomes observed in the progress tests are expected and do not negatively impact graduates. We have briefly added this information to the Discussion section (lines 335-341).

4. Were psychometric measures (e.g. difficulty, point biserial) used to evaluate test item performance? If so, were poor performing questions omitted, and is the analyzed data before or after omission?

We appreciate the question. As mentioned in previous responses and acknowledged in the limitations section of our discussion, we did not estimate psychometric measures for the administered progress tests. Therefore, these findings could not be incorporated into the analysis presented.

5. Concerns are raised by references 3 and 22. These articles report recent studies that are by some of the same authors using much of the same data. The authors should carefully review the conclusions of these earlier studies and what is novel in this the study. Reference 3 is used in the introduction instead of directly referencing early studies of progress testing. Is this choice necessary?

We appreciate the opportunity to clarify the citations mentioned.

Reference 3 (Tendencia en la retención de conocimientos de ciencias básicas en una prueba de progreso entre estudiantes de Medicina. Educ Médica. 2023;24: 100830. doi:10.1016/j.edumed.2023.100830) reports the results of a study conducted in the same setting as the present analysis; however, the research question and design were different. That study focused on evaluating the retention of knowledge in basic science courses as students progressed through the academic years. For this purpose, only data from the 2017 to 2022 progress tests were included, and the analysis was limited to responses to 45 questions specifically related to basic science courses.

On the other hand, Reference 22 (Correlación entre una evaluación sumativa escrita y el promedio ponderado en estudiantes de medicina humana. Investig En Educ Médica. 2022;11: 37–50. doi:10.22201/fm.20075057e.2022.43.22422) presents the results of a different study also conducted in the same setting. This previous research used data from the 2017 to 2020 progress tests and aimed to evaluate the correlation between scores on the progress tests and academic performance, as measured by the weighted grade point average (GPA), which is calculated based on the grades obtained in curricular courses weighted by their credit hours.

Regarding the citation of the article Tendencia en la retención de conocimientos de ciencias básicas en una prueba de progreso entre estudiantes de Medicina. Educ Médica. 2023;24: 100830. doi:10.1016/j.edumed.2023.100830, we thank the reviewer for the observation, which allowed us to identify an error in the citation. This has been corrected in the revised version, and the reference is no longer cited in the Introduction section.

Finally, we wish to confirm that this new manuscript addresses a research objective that differs from the previously cited studies. It includes broader data coverage and conducts a longitudinal trend analysis that has not been previously performed in this context.

6. Line 62 should be "students", not "student".

We corrected this error on line 65 of the current version.

Reviewer #2: The manuscript is well composed and delivers insights on an important assessment tool in medical education. The researchers have taken a sizeable sample and used reasonable statistics mostly through the research.

However, this reviewer would like to suggest some corrections or explanations to a few issues:

1. Language (translation) issues in cover page.

We have reviewed the first page of the manuscript to identify and correct any translation issues.

2. Meaning of statement in line 34-35 not clear.

We appreciate the observation and have improved the wording of that sentence in the abstract. The current version now reads: 'When stratified by entry cohort, no sustained improvement in scores was observed within cohorts over time.' (lines 33-34).

3. In line 120, the word "median score" is used in a non-statistical context. "Median" should strictly be used to describe the population and findings as one measure of central tendency.

We appreciate the observation, which allowed us to identify a wording error. We agree with the reviewer. The revised sentence now reads (line 122): 'The raw score was subsequently converted to a 20-point scale, where a score of 10 corresponds to answering 50% of the questions correctly'.

4. In line 134: Use of Pearson correlation to compare test scores raises some concerns. If we are comparing test scores of the same student in a longitudinal study, paired tests or repeated ANOVA might be more justifiable.

We appreciate the observation. In this case, we consider the choice of statistical test appropriate, as at this descriptive stage of the analysis, we do not aim to analyze paired longitudinal data. The objective of this analysis is to explore and describe the correlation between scores obtained across the eight progress tests analyzed. These are not treated as paired data; rather, at this stage, they are evaluated as independent data groups based on the year of test administration. The results of this analysis are presented in Figure 3. This observation did not result in changes to the manuscript.

5. Line 144 states that the test was somehow based on "participant identity" while line 158 says data was anonymized. It is possible to do per participant analysis in anonymized data. But the procedure should be mentioned in methodology. The statement raises ethical issues.

We appreciate the observation and agree with the reviewer. We have replaced the term 'participant identity' with a more accurate descriptor that better reflects our approach and aligns with the ethical considerations described.

The revised sentence now reads (line 146): 'The standard error was adjusted for clustering by a unique numeric identifier, with an exchangeable correlation structure.' We would also like to confirm that the analyzed database did not include any personal information that could be used to identify individual students.

6. Declining interest in PTs throughout the years is very distinctly seen. Students are taking lesser and lesser tests as years pass. This should be discussed, and reasons should be explored.

We appreciate the observation and decide to briefly describe these reasons in the second paragraph of the Discussion section. The main reason—namely, the non-mandatory nature of our progress test—was already included in that paragraph. We have now added a second possible reason for the declining number of students within each entry cohort who choose to take the progress test. This reason was reported in a previous study conducted at our university, which found that favorable perceptions of the progress test tend to decrease as students advance in their studies. A similar pattern was observed with an increasing number of progress tests taken. This could explain the differences in satisfaction levels between second- and sixth-year students when taking the test. These explanations can be reviewed in lines 291 to 305 of the revised manuscript.

7. Mean and median both used to describe data in lines 172-173. It is better to use one measure of central tendency as per normality test of the variable.

The reviewer’s observation is correct; however, we have decided to retain both measures of central tendency to prioritize a more comprehensive description of the data and to highlight the normal tendency in the distribution of this variable. Additionally, this descriptive approach allows us to present positional measures such as the first and third quartiles alongside the median. We kindly request to keep both results, as they do not affect the validity of the main findings.

8. In lines 186-189, are the years used to characterize cohort or year of test taken? As a reader, this reviewer couldn't identify it correctly. Same with Fig 3 caption. Are those years the cohort number? or the date of test?

We appreciate the reviewer’s comment, which helped us improve the clarity of the text. In the manuscript, we are referring to the year in which the progress test was administered. Therefore, eight tests are observed, corresponding to administrations from 2017 through 2024. We also took this opportunity to clarify the figure title by specifying that it refers to the year of test administration. The final version of this text can be found in lines 188 to 196 of the revised manuscript.

9. Initiation of clinical subjects in table 2 is not clear.

We appreciate the reviewer’s observation, which allowed us to use a more precise term to refer to the beginning or transition to clinical courses within the curriculum, after completion of the first three years of study. In the revised version of the manuscript, we have corrected the title of Table 2 as well as the corresponding column heading.

10. Line 215 states "required to complete", while elsewhere PTs are termed "voluntary". This confuses readers. Please try to use single narrative and explain this in methodology.

We appreciate the reviewer’s observation, which helped us improve the wording of the text to prevent confusion for the reader. Indeed, this progress test experience is characterized by its non-mandatory nature. This information is described in the subsection on study design and setting. The revised text can be found in lines 217 to 218 of the manuscript.

11. Taking the baseline score as covariate / confounder would be good during analysis so that students with better or worse baseline knowledge are accounted for. Correlation among scores of consecutive tests depends on baseline of each participant. Hence, it is better to account for it. Again, paired analysis could suit this situation better.

The reviewer’s comment is correct, and our analysis has considered this aspect from the study design stage. The primary analysis of our study focused on evaluating the trend in students’ scores over time. The longitudinal and repeated-measures nature of the data was accounted for and is described in our statistical analysis plan, from lines 144 to 156. This approach is further emphasized in the Results section, where we include a specific subsection on the longitudinal analysis of scores (line 208). The core of this analysis is the generalized estimating equations (GEE) model, with its results presented in Table 3 and Figure 6. Therefore, we believe this observation does not require changes to the manuscript.

12. One big question that this reviewer pondered about was: Why is average PT around 50%? Why is it not progressing? Is it about the modality of PT? Is it about the medical education being delivered? The authors should try to explore these findings in detail.

We thank the reviewer for this insightful observation. Indeed, our initial hypothesis anticipated a sustained increase in progress test scores over time. However, our findings must be interpreted considering the specific cha

---

## [Decision Letter · Decision Letter 1]

25 Jul 2025

Trends in progress test performance of medical students at a university in Peru

PONE-D-25-22024R1

Dear Dr. Romani,

We’re pleased to inform you that your manuscript has been judged scientifically suitable for publication and will be formally accepted for publication once it meets all outstanding technical requirements.

Kind regards,

Rosemary Bassey, Ph.D.

Academic Editor

PLOS ONE

Reviewers' comments:

Reviewer's Responses to Questions

**Comments to the Author**

1. If the authors have adequately addressed your comments raised in a previous round of review and you feel that this manuscript is now acceptable for publication, you may indicate that here to bypass the “Comments to the Author” section, enter your conflict of interest statement in the “Confidential to Editor” section, and submit your "Accept" recommendation.

Reviewer #1: All comments have been addressed

Reviewer #2: All comments have been addressed

2. Is the manuscript technically sound, and do the data support the conclusions?

Reviewer #1: Yes

Reviewer #2: Yes

3. Has the statistical analysis been performed appropriately and rigorously? 

Reviewer #1: Yes

Reviewer #2: Yes

4. Have the authors made all data underlying the findings in their manuscript fully available?

Reviewer #1: Yes

Reviewer #2: Yes

5. Is the manuscript presented in an intelligible fashion and written in standard English?

Reviewer #1: Yes

Reviewer #2: Yes

6. Review Comments to the Author

Reviewer #1: The authors have made a reasonable effort of addressing all the comments raised by reviewers. The statistics are sound, and the manuscript is solidly written in comprehensible English. One or two typographic errors remain, notably, the absence of a period on line 18, after "training". All the underlying data is easily accessible in the supplemental information. Ideally, more information should be provided on finances/funding.

Reviewer #2: The authors have taken good effort to address all the comments. The changes have been reflected in the revised manuscript and the reasoning behind some of the comments helped this reviewer learn a different perspective of the authors too. The manuscript looks publishable at this reviewer's end now.

7. PLOS authors have the option to publish the peer review history of their article (what does this mean? ). If published, this will include your full peer review and any attached files.

**Do you want your identity to be public for this peer review?** For information about this choice, including consent withdrawal, please see our Privacy Policy .

Reviewer #1: No

Reviewer #2: No

---

## [Editor Report · Acceptance letter]

PONE-D-25-22024R1

PLOS ONE

Dear Dr. Romani,

I'm pleased to inform you that your manuscript has been deemed suitable for publication in PLOS ONE. Congratulations! Your manuscript is now being handed over to our production team.

Kind regards,

on behalf of

Dr. Rosemary Bassey

Academic Editor

PLOS ONE